# Frequentist model averaging for analysis of dose–response in epidemiologic studies with complex exposure uncertainty

**Deukwoo Kwon**[1]*, **Steven L. Simon**[2], **F. Owen Hoffman**[3], **Ruth M. Pfeiffer**[4]

**1** Department of Internal Medicine, McGovern Medical School, Houston, Texas, United States of America, **2** Division of Cancer Epidemiology and Genetics, National Cancer Institute, National Institutes of Health, Bethesda, Maryland, United States of America, **3** Oak Ridge Center for Risk Analysis, Oak Ridge, Tennessee, United States of America, **4** Biostatistics Branch, Division of Cancer Epidemiology and Genetics, National Cancer Institute, National Institutes of Health, Bethesda, Maryland, United States of America

* Deukwoo.Kwon@uth.tmc.edu

## Abstract

In epidemiologic studies, association estimates of an exposure with disease outcomes are often biased when the uncertainties of exposure are ignored. Consequently, corresponding confidence intervals (CIs) will not have correct coverage. This issue is particularly problematic when exposures must be reconstructed from physical measurements, for example, for environmental or occupational radiation doses that were received by a study population for which radiation doses cannot be measured directly. To incorporate complex uncertainties in reconstructed exposures, the two-dimensional Monte Carlo (2DMC) dose estimation method has been proposed and used in various dose reconstruction efforts. The 2DMC method generates multiple exposure realizations from dosimetry models that incorporate various sources of errors to reflect the uncertainty of the dose distribution as well as the uncertainties in individual doses in the exposed population. Traditional measurement-error model approaches, typically based on using mean doses in the dose-exposure analysis, do not fully account exposure uncertainties. A recently developed statistical approach that overcomes many of these limitations by analyzing multiple exposure realizations in relation to disease risk is Bayesian model averaging (BMA). The analytic advantage of the BMA is its ability to better accommodate complex exposure uncertainty in the risk estimation, but a practical. Drawback is its significant computational complexity. In this present paper, we propose a novel frequentist model averaging (FMA) approach which has all the analytical advantages of the BMA method but is much simpler to implement and computationally faster. We show in simulations that, like BMA, FMA yields 95% confidence intervals for association parameters that close to 95% coverage rate. In simulations, the FMA has shorter length of CIs than those of another frequentist approach, the corrected information matrix (CIM) method. We illustrate the similarities in performance of BMA and FMA from a study of exposures from radioactive fallout in Kazakhstan.

**Data Availability Statement:** The data used in the simulations are reconstructed radiation doses from fallout from nuclear testing at the Semipalatinsk Nuclear Test Site in Kazakhstan (Land CE, et al.

Accounting for shared and unshared dosimetric uncertainties in the dose-response for ultrasound-detected thyroid nodules following exposure to radioactive fallout. Rad Res. 2015; 183:159-173). These data are subject to third party restrictions and while we are authorized to use them for this publication we cannot release them. Interested researchers need to apply (as the authors have) for permission to use the data from the National Cancer Institute, NIH, HHS. Contact person: Todd Gibson todd.gibson@nih.gov

**Funding:** DK was partially supported by the Biostatistics/ Epidemiology/ Research Design (BERD) component of the Center for Clinical and Translational Sciences (CCTS) for this project that is currently funded through a grant (UL1TR003167), funded by the National Center for Advancing Translational Sciences (NCATS), awarded to the University of Texas Health Science Center at Houston. RMP was partially supported by the Intramural Research Program of the National Cancer Institute. The funders had no role in study design, data collection and analysis, decision to publish, or preparation of the manuscript.

**Competing interests:** The authors have declared that no competing interests exist.

## Introduction

Traditional statistical methods to accommodate exposure uncertainty when estimating associations of the exposure with an outcome (e.g., disease status), such as regression calibration and the Simulation Extrapolation (SIMEX) method use a single measurement or estimate of the exposure per individual and then correct the bias in the association estimates analytically or via simulations, based on assumptions about the measurement error structure [1, 2]. More recently, exposure uncertainties have been characterized by generating multiple exposure realizations (referred to herein as "cohort dose vectors") for the entire exposed population, using a strategy that incorporates any sources of random and systematic uncertainties that can be characterized in estimating the exposure. Important examples that use such exposure reconstruction are studies of cancer risk, where the exposure, i.e., the radiation dose received by specific organs in individuals, is obtained from complex dosimetry systems that also consider different components and sources of exposure, and dose estimation error. Statistical methods proposed in the literature for estimating association parameters in a dose-response relationship using multiple exposure realizations include Bayesian model averaging (BMA) [3], a frequentist Corrected Information Matrix approach (CIM) [4], Monte Carlo maximum likelihood (MCML) [5], and multiple imputations for measurement-error correction (MIME) [6]. The drawback of the BMA and MCML methods is their substantial computational burden. Additionally, MCML cannot be readily extended to more than one parameter in the dose-response function since parameter estimation is based on a grid search over the parameter space, which further increases numerical and computational complexities. CIM also is computationally expensive for large datasets as it requires inverting the variance-covariance matrix of the model, and can only be applied to a limited set of statistical models for continuous and count outcomes, but currently does not accommodate binary outcomes. Moreover, CIM implicitly assumes the single cohort vector of individual mean doses obtained by averaging the individual doses across all dose realizations per person is an unbiased estimate of the true dose–an assumption that is fundamentally incompatible with the design of the 2DMC when much of the uncertainty in estimates of the true, unknown individual doses is due to numerous sources of systematic and/or shared errors in the dose estimation.

In this article we propose and study frequentist model averaging (FMA), a novel, computationally efficient method to accommodate uncertainty in the dose-response parameter estimates when analyzing the dose response relationship with multiple exposure realizations. As it is easy to implement, FMA provides a viable alternative to BMA, which, despite all its advantages, requires expertise in Bayesian computation for efficient implementation.

We apply FMA to data from a study that aimed to assess the association of exposure to radioactive fallout from nuclear testing conducted in Kazakhstan with the risk of thyroid nodules in the local population [7]. We also compare FMA to CIM using simulations to examine the properties of the respective statistical estimates and their corresponding confidence intervals for different magnitudes and sources of exposure uncertainty. In the Kazakhstan data example we compare the results from FMA to previously published results obtained using BMA.

The remainder of this article is organized as follows: In Section 2 we briefly describe the strategy for assessment of exposures, its uncertainty components and the resulting multiple realizations of cohort vectors of individual doses. We then present the statistical model and dose response estimation methods, BMA and the novel FMA. In Section 3, we show the comparability of BMA and FMA using radiation dose and disease data obtained for the Kazakhstan cohort [7]. In Section 4, we compare FMA with CIM in a series of simulations with increasing

amounts of systematic and/or shared sources of uncertainty in dose estimation. Section 5 concludes with a discussion.

## Methods

Here we first discuss the importance of using multiple exposure realizations in epidemiologic association studies with complex exposure uncertainties and summarize a method for generating these realizations using a two-dimensional Monte Carlo (2DMC) algorithm [8]. The 2DMC is useful when there are complex uncertainties in exposures comprised of various, and often differing, degrees of systematic (shared) and random (unshared) errors for members of an epidemiologic cohort. We use the terms "dose" and "exposure" interchangeably throughout this article to refer to radiation dose, but exposure can be derived in other contexts for chemicals or other contaminants.

### Exposure assessment

Epidemiologic studies to determine the dose-response association for individuals exposed to radiation, chemical agents, or toxins, have numerous challenges, some of the most important and difficult ones arising from the uncertainty in the exposure estimates for the study participants.

Exposure assessment is a well-developed field and encompasses many methods and strategies depending on the exposing agent (e.g., radiation). Exposure assessment methods can use measurement data on concentrations of the toxin in the environment. However, typically, such information is available only on a group-average or location-average basis and rarely for individual members of a cohort. For that reason, exposure assessment in epidemiologic studies often relies on estimation of individual exposures from mathematical models using appropriate input data to characterize the exposure conditions for each study participant. Exposure assessment models vary widely in their level of complexity, ranging from simple correlation models based on observations between the amount of toxin or radiation absorbed by the body per unit of toxin or radiation in the environment, to detailed mechanistic models that attempt to describe the physical and biological processes leading to exposure and dose. In radiation epidemiology, where estimation of individual exposures and the estimation of exposure-related uncertainties for each individual dose have become routine practice, most recently Monte Carlo methods have been used [8–10].

### Shared and unshared errors and generation of multiple exposure realizations using the two-Dimensional Monte Carlo (2DMC) approach

In mathematical models used to reconstruct individual doses to study participants, some parameters have an unknown true, fixed value that is shared among all members of a defined subgroup, e.g., cohort subjects of each gender or within an age group, or residents of a specific town or village. For the members of such a subgroup, this lack of knowledge of the true value is a systematic and/or shared source of uncertainty equally affecting all estimates of dose for those persons. In contrast, some parameters can be treated as varying independently among members of a subgroup and/or between subgroups and are a source of unshared error. Parameter values that are specific to an individual and independent from values specific to other persons are sources of unshared errors.

The importance of shared errors is that they are a source of systematic error among the subgroup sharing the estimated but uncertain parameter value. Systematic errors are a primary contributor to bias in the estimate of the dose response in epidemiological studies. The two-Dimensional Monte Carlo (2DMC) approach was developed to derive numerous possible

versions of the unknown true exposure by simulating alternative values for systematic errors as well as unshared errors [8]. Each set of study doses, termed "realization" or "cohort dose vectors", contains a dose estimate for each individual in the study and is based on a specific selection of shared and unshared parameter values.

Shared parameters are sometimes characterized by discrete distributions, whereby each randomly selected value can lead to dramatically different numerical estimates of the exposure. Because the probability distributions from which shared parameter values are selected cannot be guaranteed to reflect the true dose generating processes, the average value of the generated dose realizations must not be interpreted as an unbiased estimate of the mean value of the exposures to the study participants.

To characterize the many possible alternative combinations of shared and unshared errors, hundreds to thousands of alternative realizations of cohort dose vectors are generally created for dose-response analyses. These multiple realizations are created independently of one another and could, in theory, be developed from different dose-related data or even from different exposure assessment models. There are no limits on the types of data or on the combinations of exposure assessment models that might be used to generate multiple realizations for a dose-response analysis. The purpose of using different dose assessment algorithms to generate multiple realizations is to provide numerous alternatives of possibly true conditions of shared and unshared errors in dose estimation.

The 2DMC approach has been used in several cohort-based dose reconstructions in radiation epidemiology, including fallout exposures from nuclear testing in Kazakhstan [7], occupational radiation exposure among U.S. medical radiation technologists [11] and pediatric patients exposed to computerized tomographic imaging (CT) [12]. In contrast, there is a very simplistic strategy for simulating multiple realizations of cohort dose vectors known as the SUMA ("Shared and Unshared, Multiplicative and Additive") model [9] that we describe in more detail in Section 4 and use to test FMA against CIM.

## Statistical models for the association of exposure with disease risk

The main goal of dose-response analysis is to estimate the association parameter that relates exposure to risk of a disease outcome, and the corresponding 95% confidence or credibility interval (CI), accounting for various levels of uncertainty in the exposure estimation. For illustration, we use radiation absorbed dose expressed in units of Gray (Gy) to represent exposure.

Statistical models for risk estimation due to radiation exposure are described in the BEIR VII Report [13]. In cohort studies of cancer, the quantity of interest typically is the incidence rate, modeled as a function of radiation dose, age at exposure, gender, and other covariates [14]. Incidence is often estimated from grouped survival outcome data using Poisson regression models. Two popular models for the incidence rate as a function of radiation dose $D$ are the

1. Excess Relative Risk (ERR) model: $\lambda_0[1 + ERR(D)]$, and the

2. Excess Additive Risk (EAR) model: $\lambda_0 + EAR(D)$,

where $\lambda_0$ denotes the baseline incidence rate at zero dose, and $ERR(D)$ and $EAR(D)$ the additional risk conveyed by exposure. Both $ERR(D)$ and $EAR(D)$ can be modeled using many different functional forms, including, but not limited to, linear ($\beta * dose$) and linear-quadratic ($\beta * dose + \gamma * dose^2$) functions. Individual level (ungrouped) survival outcome data can be analyzed using proportional hazards regression models, similar to the standard Cox regression model. Ignoring time to event information and only using binary outcomes, some authors used logistic regression models to analyze cohort studies [7, 15, 16]. In a logistic model the

excess odds ratio (EOR) is used instead of the excess relative risk (ERR). In this article we focus on ERR and EOR modeling but extensions to the EAR model are straight forward.

## Model averaging methods: BMA and FMA

Model averaging methods have not been widely used in dose-response analysis applications although there are numerous statistical publications describing their form and usage, e.g. [17–20]. Implicit in all model-averaging strategies is weighting various dose-response models by their goodness of fit to the observed data, i.e., their plausibility in explaining the relationship. Model averaging approaches thus are very well suited to analyzing multiple realizations of possible cohort doses from the 2DMC method.

For both BMA and FMA, first a probability model that relates disease status to exposure is specified. In such a model we let $D$ denote exposure, $C$ denote confounding variables and $M$ possible effect modifying variables. For a binary disease outcome $Y$, with $Y = 1$ for event and $Y = 0$ otherwise, the probability model is typically logistic,

$$\frac{p_i}{1 - p_i} = \exp(\alpha_0 + \alpha_1 C_i)[1 + \beta D_i\{\exp(\eta M_i)\}], \tag{1}$$

where $p_i = P(Y_i = 1)$, and $\beta$ is the EOR per unit exposure, $D$. Letting $\Theta = (\alpha_0, \alpha_1, \beta, \eta)$, the corresponding likelihood function is

$$L(\Theta) = \prod_{i=1}^{N} p_i^{y_i}(1 - p_i)^{(1-y_i)}. \tag{2}$$

For individual-level survival data $(T, \delta)$, where $T$ denotes the event time and $\delta$ the censoring indicator, that is one if the event occurs and zero otherwise, a proportional hazards model can be written as

$$h(t) = h_0(t)\exp(\alpha_0 + \alpha_1 C_i)[1 + \beta D_i\{\exp(\eta M_i)\}], \tag{3}$$

where $h_0(t)$ denotes the baseline hazard function and $\beta$ is the ERR per unit of exposure, $D$. The parameters $\Theta$ can be estimated from the partial likelihood function,

$$L(\Theta) = \prod_{I \in \Lambda} \frac{\exp(\alpha_0 + \alpha_1 C_i)[1 + \beta D_i\{\exp(\eta M_i)\}]}{\sum_{k \in R(t_i)}\exp(\alpha_0 + \alpha_1 C_i)[1 + \beta D_i\{\exp(\eta M_i)\}]}, \tag{4}$$

where $\Lambda$ denotes the set of event times, $R(t)$ denotes the set of individuals still at risk in the cohort at time t.

For survival outcomes grouped into strata defined by multiple variables, the number of events in stratum i, $Y_i$, has a Poisson distribution,

$$Y_i \sim Poisson(\exp(\alpha_0 + \alpha_1 C_i)[1 + \beta D_i\{\exp(\eta M_i)\}] \times PY_i), \tag{5}$$

where $PY_i$ denotes the person-years for stratum i. The corresponding log-likelihood function is

$$L(\Theta) = \sum_i\{Y_i(\log PY_i + \alpha_0 + \alpha_1 C_i + \log(1 + \beta D_i\{\exp(\eta M_i)\}))\} - \sum_i\{PY_i \exp[\alpha_0 + \alpha_1 C_i]([1 + \beta D_i\{\exp(\eta M_i)\}])\}$$

$$- \sum_i \log(Y_i!). \tag{6}$$

However, in studies with reconstructed dose, instead of the true dose $D$, which is unobserved, we have $K$ realizations $X_k$, $k = 1, \ldots, K$, of the cohort dose vector (e.g. K = 5,000), that were generated by the 2DMC algorithm described in the previous Section. Thus the data

available for the $i^{th}$ individual in the cohort are $(Y_i, PY_i, X_{i1}, \ldots, X_{iK}, C_i, M_i)$. In all models described here, (1), (3), and (5), the quantity of interest is the dose-response parameter, β.

We next briefly present two methods for incorporating multiple dose realizations into parameter estimation of the dose-response model and uncertainty quantification, the previously published BMA and the novel FMA methods.

## Bayesian model averaging (BMA)

We illustrate the BMA using the logistic model in (1) and omit effect modifying variables, $M$, for simplicity. The BMA method, described in detail in [3] that we employ in this paper, obtains the posterior probability of the ERR dose-response parameter β in Eq (1) based on using each dose realization in the likelihood (2) weighted by the goodness-of-fit of the corresponding model to the vector of disease outcomes among the individuals in the cohort. As a measure of uncertainty, a credibility interval based on the posterior distribution for all dose-response parameters is computed. In brief, in addition to the parameters $\Theta = (\alpha_0, \alpha_1, \beta)$ in model (1), we introduce the dose index parameter, γ, where $\gamma = i$ corresponds to the $i^{th}$ dose realization $X_i$ being used in the likelihood (2). Thus, the joint posterior distribution of all parameters is given by

$$p(\alpha_0, \alpha_1, \beta, \gamma \mid \underline{Y}, \underline{X}, \underline{C}) \propto l(\alpha_0, \alpha_1, \beta, \gamma \mid \underline{Y}, \underline{X}, \underline{C})p(\alpha_0)p(\alpha_1)p(\beta)p(\gamma)p(\underline{\pi}). \qquad (7)$$

Underlined letters, $\underline{Y}, \underline{X}$, and $\underline{C}$ denote data vectors for the whole cohort. Weakly informative prior distributions $p(\alpha_0), p(\alpha_1)$ and $p(\beta)$ are used, for example normal distributions with large variances such as $N(0, 1,000)$, and the prior distribution for $\gamma \epsilon \{1, \ldots, K\}$ is likewise a only weakly informative multinomial distribution, *multinomial* $(\underline{\pi})$, where K denotes the number of individual dose realizations, and the hyperprior for γ, $\underline{\pi} = (\pi_1, \ldots, \pi_K)$ is *Dirichlet*$(1, \ldots, 1)$. For these choices of priors and hyperpriors every dose realization has a priori equal weight 1/K and thus every realization is equally likely to give the best fit to the data. Using weakly informative or non-informative priors in the Bayesian analysis reduces the impact of the prior on the inference and thus maximizes the influence of data. The weakly informative prior for the dose index parameter, γ, reflects our lack of knowledge about which dose realization is closest to the true unknown dose, and sets all dose realization to be equally likely to be closest to the true dose. As γ is a categorical variable, its prior is usually a Dirichlet prior with parameters all equal to one. The posterior distribution for the parameters can be obtained from a Markov Chain Monte Carlo (MCMC) algorithm using the Metropolis-Hastings (MH) method [21].

We also compute a Bayesian weight (BW$_k$), namely the relative selection frequency for a specific realization of a cohort dose vector $X_k$,

$$BW_k = \frac{p(\gamma = k \mid y, C, X_k)}{\sum_{m=1}^{K} p(\gamma = m \mid y, C, X_m)}, k = 1, \ldots, K, \text{ where } 0 \leq BW_k \leq 1, \qquad (8)$$

where $p(\gamma = k \mid y, C, X_k) = \iiint p(\alpha_0, \alpha_1, \beta, \gamma = k \mid y, C, X_k)d\alpha_0 d\alpha_1 d\beta$ is the marginal posterior distribution of the $k^{th}$ cohort dose realization. $BW_k$ in Eq (8) is a relative measure of the goodness-of-fit of the $k^{th}$ cohort dose vector realization $X_k$ to the data, with larger values corresponding to a better fit of $X_k$ to the observed disease outcomes.

The Bayesian credibility interval for the BMA estimate of β is obtained from the highest posterior density (HPD) interval given a specific credible level (e.g., 95%). Unlike the currently available version of the CIM approach, BMA can handle any dose-response model, regardless

of whether the disease outcomes are expressed as continuous, binary, count, or survival data. For further details about BMA see [3].

## Frequentist model averaging (FMA)

A drawback of the BMA approach is that it is extremely expensive computationally with large dataset. In implementing the BMA, the whole dataset has to be read into memory and a large model space needs to be evaluated to obtain posterior model probabilities. For example, it takes several days on a personal computer with a multicore processor and 24GB memory to get the BMA result for the example data in this paper using traditional Bayesian computational softwares such as **WinBUGS** [22] and **JAGS** [23]. When we have complex uncertainties, conventional Markov chain Monte Carlo (MCMC) methods such as Gibbs sampling and the Metropolis-Hastings algorithm face the 'local trap' problem due to many local optima of the posterior distribution of β. Advanced Bayesian computation techniques are then needed to get a proper posterior distribution of β, which requires special programming expertise to perform the analysis. To avoid the computational burden of BMA, we, therefore, propose a new frequentist model averaging (FMA) approach for estimating the ERR(D), β, and its corresponding 95% confidence interval.

Before describing our specific FMA approach, we provide a general description of the FMA idea, see also [24]. In the FMA approach, distinct models fit to the data are averaged. "Different models" could refer to different parametric forms or models including different sets of independent variables. For example, assume that there are three independent variables $(X_1, X_2, X_3)$ available for inclusion in a linear regression model, leading to seven different possible models (Model 1 $(M_1)$: $X_1$ is only included; Model 2 $(M_2)$: $X_2$ is only included; Model 3 $(M_3)$: $X_3$ is only included; Model 4 $(M_4)$: $X_1$ and $X_2$ are included; Model 5 $(M_5)$: $X_1$ and $X_3$ are included; Model 6 $(M_6)$: $X_2$ and $X_3$ are included; and Model 7 $(M_7)$: $X_1$, $X_2$, and $X_3$ are included) assuming at least one independent variable is significant. The FMA estimates of the regression coefficients are then obtained as the weighted average of the coefficient estimated coefficients of the different models, using model weights (MW) based on a goodness-of-fit (GOF) criterion [25], e.g. the Akaike Information Criterion (AIC), defined as $AIC = 2p - 2\ln(\hat{L})$, where $p$ denotes the number of estimated parameters in the model and $\hat{L}$ denotes the maximum value of the model's likelihood function],

$$\hat{\boldsymbol{\beta}}_{FMA} = \sum_{k=1}^{7} MW_k\left(\hat{\boldsymbol{\beta}}_{M_k}\right), \tag{9}$$

where $MW_k = \dfrac{exp\left(\frac{1}{2}AIC_k\right)}{\sum_{m=1}^{K} exp\left(\frac{1}{2}AIC_m\right)}, k = 1,\dots,7$ and $\hat{\boldsymbol{\beta}}_{M_k}$ denotes the coefficient estimator of Model $k$. FMA for generalized linear models (GLMs) and generalized linear mixed models (GLMMs) were discussed, for example, in [26, 27] and FMA for high-dimensional data was considered in [28, 29].

Here we present FMA in a special setting, since all models have the exact same parametric form (i.e., the same parametric regression model with the same number of parameters) unlike the example described above. The difference between the models is the specific exposure vector that is used as the independent variable. The corresponding model weights $MW_k$ can be viewed as corresponding to the posterior probabilities of the parameters γ = k, k = 1,...,K in BMA. However, in contrast to the BMA method that simultaneously evaluates all dose realizations, our FMA strategy uses a two-stage approach. In the first stage, each realization of a cohort dose vector, $X_k$, k = 1,..,K, along with the vector of disease outcomes and associated

confounding variables is evaluated separately using the same parametric model form. The resulting $K$ models can be easily fit in parallel runs, unlike BMA which evaluates all dose vectors jointly and for which computations cannot be parallelized. As FMA allows for parallel implementations there are huge savings in computational time when the sample size, N, and the number of exposure realizations, K, are large. To run BMA, the whole dataset has to be read into computer memory space, which typically prevents BMA analyses on standard single computers with large datasets. Each model yields an estimate of the ERR(D), $\hat{\beta}_k$, the corresponding variance, $var\left(\hat{\beta}_k\right)$, and GOF measure. Here we use the AIC [24] to compute a frequentist weight,

$$FW_k = \frac{exp\left(\frac{1}{2}AIC_k\right)}{\sum_{m=1}^{K} exp\left(\frac{1}{2}AIC_m\right)}, k = 1, \ldots, K, \text{ where } 0 \leq FW_k \leq 1. \qquad (10)$$

In stage 2, given $FW_k$, we then simulate $n_k = FW_k \times M$ values $\hat{\beta}_k(s)$ from the normal density $N\left(\hat{\beta}_k, var\left(\hat{\beta}_k\right)\right)$, for each $k = 1, \ldots, K$ and an arbitrarily large $M$ (e.g., $M = 100,000$). From this setup, we obtain M samples of $\hat{\beta}$. The final FMA estimate $\hat{\beta}_{FMA}$, and its corresponding 95% confidence interval are obtained as the weighted mean of the M simulated $\hat{\beta}$ values, and the 2.5th and 97.5th percentiles of their corresponding empirical distribution function. The FMA approach is summarized below in Fig 1.

## Simulations and data example to illustrate FMA and its comparability with BMA

We use data from the study by Kwon et al. [3] to further illustrate the FMA approach and compare it to results obtained from the BMA method. The study cohort consists of 2,376 individuals under the age of 21 years between August 1949 and September 1962, who lived in villages in the downwind area of the Semipalatinsk Nuclear Test Site located in northeast Kazakhstan. Radiation dose estimates for individuals were based on standard dose reconstruction models (NCRP 2009) while the multiple realizations were derived using the 2DMC approach [8] supplemented with probability distributions describing interview data on exposure-related lifestyle habits [30]. Using the 2DMC, 5,000 distinct radiation exposure realization vectors were generated for this analysis. The aim of the analysis was to estimate the association of reconstructed fallout-related internal and external radiation doses to the thyroid gland and the prevalence of thyroid nodules (absent/present) using a logistic regression model. We analyzed the binary outcome $Y = 0 \text{ or } 1$ (thyroid nodules absent or present) with $p_i = P(Y_i = 1)$ modeled using the logistic model $\frac{p_i}{1-p_i} = \exp(\alpha_0 + \alpha_1 C_i)[1 + \beta X_i\{\exp(\eta M_i)\}]$, where $\beta$ is the excess odds ratio per Gy (EOR).

## Comparing FMA and BMA using simulated settings

First, we compared BMA and FMA using the 360 settings (3 different $\beta$ values × 2 uncertainty scenarios × 60 test disease outcome vectors) examined in the simulation study in [3]. In brief, we simulated multiple alternative realizations of a "true" set of disease status variables (i.e., the disease status for the 2,376 individuals in the cohort). Each simulated set of disease status variables was produced using one of the 5,000 vectors of doses for the cohort (subsequently removed from the multiple realizations, leaving 4,999 for testing), one of four pre-specified values of a "true" slope, $\beta$, (EOR/Gy = 0, 3, 12, and 20), and covariates that affect the baseline risk of thyroid nodules (i.e., age at time of screening and sex). We used the following operating

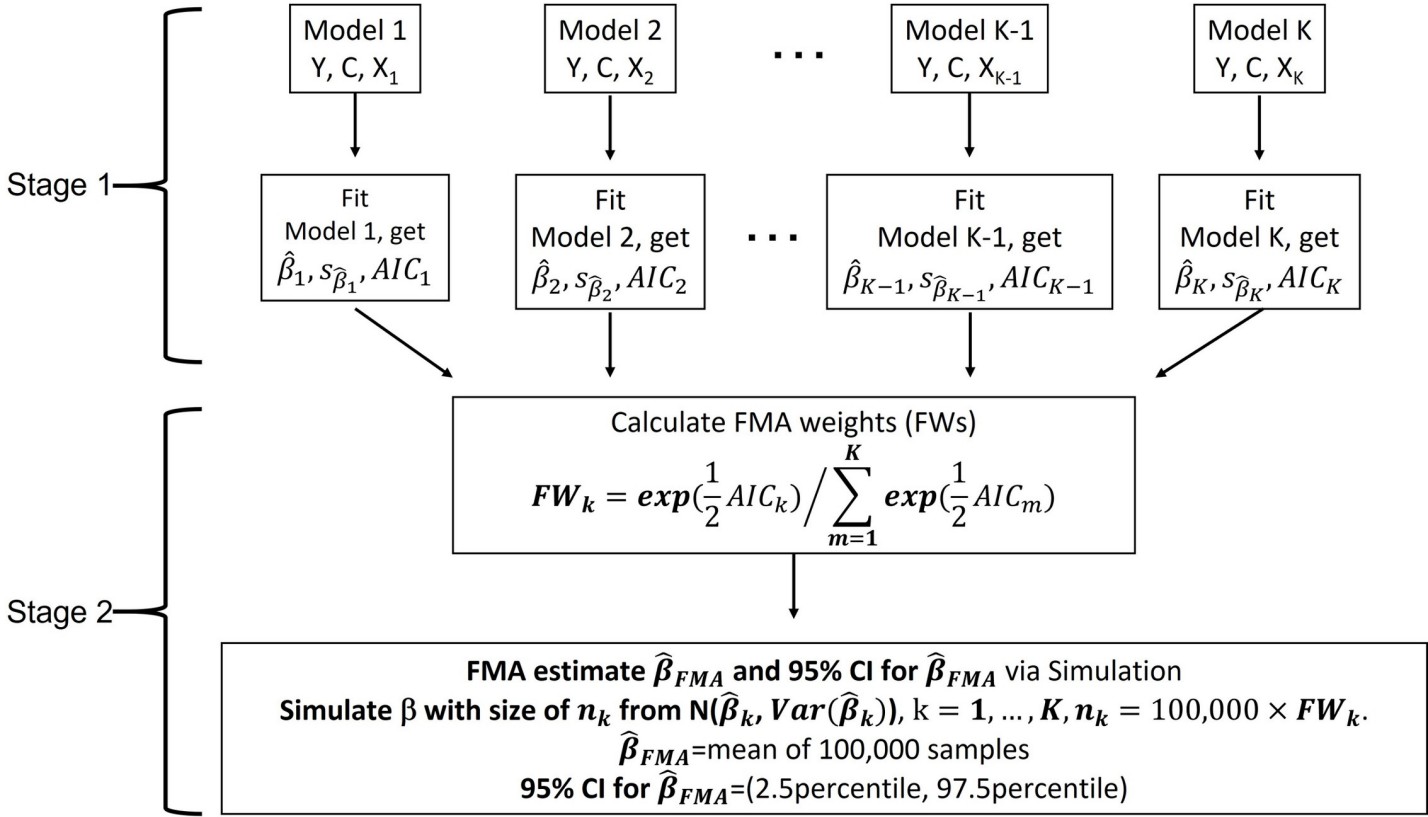

**Fig 1. Schema of simulation-based frequentist model averaging (FMA) approach.** GOF stands for goodness-of-fit measure. AIC stands for Akaike information criterion, defined as $AIC = 2p - 2 \ln(\hat{L})$, where $p$ denotes the number of estimated parameters in the model and $\hat{L}$ denotes the maximum value of the likelihood function for the model.

characteristics to assess performance of the methods: the coverage rate of the 95% confidence intervals (CR), the left error rate (LER), the right error rate (RER), the length of the 95% confidence interval (LCI), the absolute relative bias of EOR (ARB), and the relative upper limit (RUL). CR, RER, and LER were defined as the frequency (percentage of times) the true value of parameter was inside the CI, below the lower limit, and above the upper limit, respectively. ARB was calculated as $ARB = abs(\hat{\beta}_{FMA(BMA)} - \beta)/ \beta$, where $\beta$ denotes the true test value of $\beta$. RUL was calculated as the ratio of the upper limit of the 95% CI and $\hat{\beta}_{FMA}$ $\hat{\beta}_{BMA}$). The performance measures for FMA and BMA are shown in Table 1.

In settings with small systematic uncertainties, as was the case for doses only describing external exposure to fallout deposited on the ground surface, both BMA and FMA yielded almost identical results in terms of CR, ARB, LCI and RUL (Table 1). In settings with substantially larger uncertainty, FMA showed slightly higher coverage rates for the two larger test values (EOR = 12 and 20). These higher coverage rates of FMA are due to the longer length of the FMA 95% CIs. However, for EOR = 3, FMA had a similar performance to that of BMA. To compare the efficiency of BMA and FMA, we present the averages of their respective coverage rates, which were very similar, about 97.1% (BMA) and 98.4% (FMA) for the setting of small uncertainty; and 95.8% (BMA) and 96.7% (FMA) for the setting of large uncertainty. In Fig 2A and 2B FMA shows quite comparable weighted mean estimates of EOR as BMA and the respective confidence/credible intervals are also similar.

**Table 1. Summary of comparison between BMA and FMA for 240 simulated test data sets.**

| | | Coverage Rate (%) (LER, RER) | | ARB (95% CI) | | LCI (95% CI) | | RUL (95% CI) | |
|---|---|---|---|---|---|---|---|---|---|
| Small uncertainty | True EOR/Gy | BMA* | FMA | BMA* | FMA | BMA* | FMA | BMA* | FMA |
| | 20 | 96.7 (1.65, 1.65) | 96.7 (1.65, 1.65) | 0.12 (0.004, 0.510) | 0.12 (0.0004, 0.486) | 16.70 (12.22, 21.75) | 16.60 (13.39, 22.54) | 1.42 (1.40, 1.465) | 1.42 (1.374, 1.461) |
| | 12 | 96.7 (1.65, 1.65) | 96.7 (1.65, 1.65) | 0.12 (.008, 0.467) | 0.13 (0.004, 0.426) | 10.70 (8.047, 14.03) | 10.62 (8.342, 14.40) | 1.46 (1.428, 1.500) | 1.46 (4.412, 1.512) |
| | 3 | 100 (0 0) | 100 (0, 0) | 0.19 (0.016, 0.563) | 0.17 (0.019, 0.549) | 4.70 (3.576, 5.534) | 4.72 (3.927, 5.761) | 1.75 (1.636, 1.992) | 1.76 (1.624, 2.046) |
| | 0 | 95 (2.5, 2.5) | 100 (0, 0) | 0.66[#] (0.028, 1.763) | 0.37[#] (0.028, 1.089) | 4.14 (2.565, 6.22) | 4.45 (3.276, 5.585) | 2.47[^] (0.068, 5.166) | 2.21[^] (0.836, 3.920) |
| Large uncertainty | EOR/Gy | BMA* | FMA | BMA* | FMA | BMA* | FMA | BMA* | FMA |
| | 20 | 93.3 (5.05, 1.65) | 96.7 (3.3, 0) | 0.25 (0.006, 0.838) | 0.24 (0.018, 0.725) | 26.44 (11.62, 42.77) | 29.0 (17.09, 56.55) | 1.7 (1.601, 1.868) | 1.68 (1.514, 1.863) |
| | 12 | 93.3 (6.7, 0) | 93.3 (6.7, 0) | 0.24 (0.024, 0.775) | 0.24 (0.056, 0.862) | 16.58 (6.702, 26.27) | 17.83 (9.853, 27.82) | 1.76 (1.623, 2.058) | 1.71 (1.532, 1.984) |
| | 3 | 98.3 (1.7, 0) | 98.3 (0, 1.7) | 0.27 (0.019, 0.745) | 0.29 (0.015, 0.812) | 5.60 (2.319, 8.823) | 5.95 (3.267, 9.593) | 2.01 (1.793, 2.536) | 1.93 (1.656, 2.435) |
| | 0 | 98.3 (1.7, 0) | 98.3 (0, 1.7) | 0.31[#] (0.022, 1.011) | 0.26[#] (0.016, 0.747) | 2.08 (1.313, 3.261) | 1.82 (1.044, 2.862) | 1.36[^] (0.162, 3.153) | 1.01[^] (0.06, 2.24) |

*BMA results were reproduced from Kwon et al. 2016. For each EOR/Gy, 60 tests were performed. # denotes average of absolute bias and ^ denotes average of upper limits.

## Comparing FMA and BMA using the observed cohort data

Table 2 shows results obtained from applying the risk evaluation strategies from two models described in [7] using the same 2DMC dataset for uncertain doses and the actual disease vector of cohort members with and without thyroid nodules. In this evaluation of the unknown slope of a linear dose response relationship, Model 1 (external and internal dose) describes the (EOR) separately in men and women and Model 2 (external dose only) describes the slope between external and internal exposures for both sexes combined.

In the actual epidemiological investigation [7], unlike in the simulated situations in Table 1, the true value of the EOR(D) was unknown. The central estimates of EOR(D) from FMA were slightly smaller than those from BMA. The upper limits of the 95% CIs from both BMA and FMA were similar. However, unlike BMA, the lower limits from FMA were near zero. The discrepancy in the lower limits arises because BMA incorporates a non-negativity constraint for the EOR estimate by using the exponential distribution as a prior distribution of β.

## Simulation study comparing FMA with CIM

We further compared the performance of the FMA method with the Corrected Information Matrix approach (CIM) [4], another frequentist approach to accommodate exposure uncertainty in the estimation of CIs. The CIM approach is described in detail in the Appendix. As currently the CIM only accommodates linear and Poisson regression models, we could not include it in the comparisons based on the Kazakhstan cohort.

## Data generation

Here, we simulated cohort disease vectors from a Poisson regression model, i.e., we assumed that the count outcome, $Y_i$, for stratum $i$ had a Poisson distribution shown in Eq (3). We used

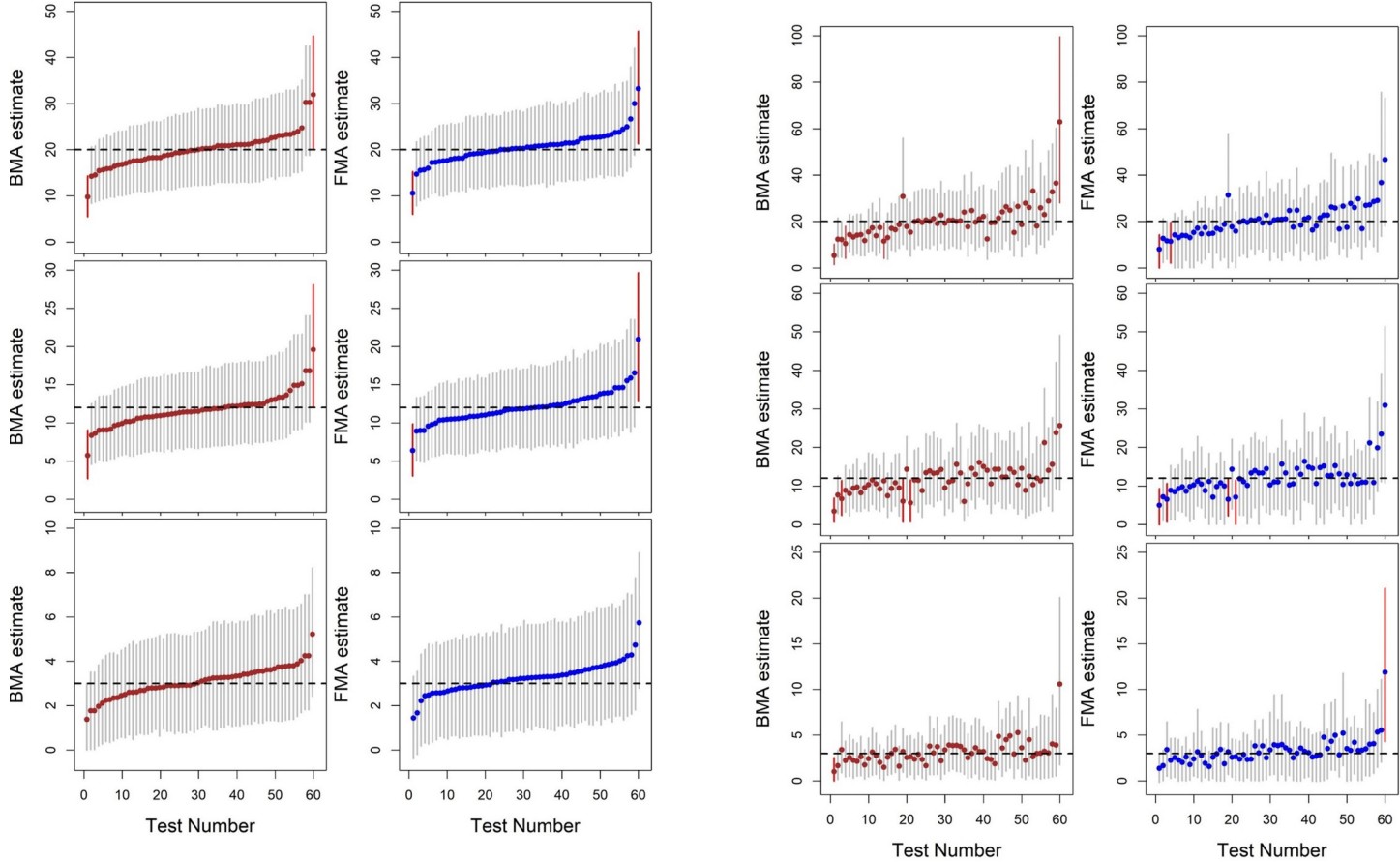

**Fig 2.** A. Comparison of BMA (left column) and FMA (right column) for external KZ dose. The red vertical line denotes 95%CI does not include the true value. The dashed horizontal line denotes the value of true $EOR_{Gy}$ (= 20, 12, and 3). B. Comparison of BMA (left column) and FMA (right column) for external KZ dose. The red vertical line denotes 95%CI does not include the true value. Dashed horizontal line denotes the value of true $EOR_{Gy}$ (= 20, 12, and 3).

scalar confounding and effect modification variables here and denote the vector of all parameters by $\Theta = (\alpha_0, \boldsymbol{\alpha_1}, \beta, \boldsymbol{\eta})$. For a cohort of N individuals the true disease generating data are $(Y_i, PY_i, D_i, \boldsymbol{X_i}, C_i, M_i)$, $i = 1, \ldots, N$. Letting $\underline{Y}, PY, \underline{C}, \underline{M}$, and $\underline{D}$ denote the vectors of count outcomes, person-years, confounder, effect modifier, and exposure, respectively, for the whole cohort, the corresponding log-likelihood is shown in the Eq (5).

**Table 2. Comparison of BMA and FMA analyses for EOR/Gy for males vs. females using total dose (external + internal) and external vs. internal dose (genders combined).** Results are for the specific exposure conditions of the Kazakhstan cohort (Land et al.2015).

| Model | Parameter | Parameter description | EOR/Gy Estimate (95% CI) | |
|---|---|---|---|---|
| | | | BMA* | FMA |
| Model 1** | β1 | Total dose (external + internal) for males (high dose uncertainty) | 9.99 (2.33, 19.1) | 8.88 (0.38, 18.3) |
| | β2 | Total dose (external + internal) for females (high dose uncertainty) | 0.35 (0.00001, 1) | 0.46 (0.0003, 1.45) |
| Model 2** | β3 | External dose for males and females combined (low dose uncertainty) | 1.47 (0.00004, 3.74) | 1.67 (0.00005, 3.78) |
| | β4 | Internal dose for males and females combined (high dose uncertainty) | 3.59 (0.11, 9.73) | 2.66 (0.0002, 7.21) |

*BMA results were reproduced from Kwon et al. 2016.

**Model 1 and Model 2 as described in Land et al. (2015)

To efficiently generate exposure vectors, we used the shared uncertainty multiplicative/additive (SUMA) error model [9] as a simplistic replacement for a complex dosimetry system. SUMA assumes that the distribution of the true dose, $X_i$, has four error components, a shared multiplicative error ($\varepsilon_{SM}$), an unshared multiplicative error ($\varepsilon_{UM,i}$), a shared additive error ($\varepsilon_{SA}$), and an unshared additive error ($\varepsilon_{UA,i}$). The multiplicative errors are assumed to be log-normally distributed and additive errors are assumed to be normally distributed. The dose $X_i$, given the errors and the mean dose, $Z_i$, is defined as

$$X_i = \varepsilon_{SM}\varepsilon_{UM,i}Z_i + \varepsilon_{SA} + \varepsilon_{UA,i}, \tag{11}$$

where $E(\varepsilon_{SM}) = E(\varepsilon_{UM}) = 1$ and $E(\varepsilon_{SA}) = E(\varepsilon_{UA}) = 0$ where $S$, $U$, $M$, and $A$ indicate shared, unshared, multiplicative, and additive errors, respectively. In our simulations we only considered shared and unshared multiplicative errors in (8) and generated $X_i = \varepsilon_{SM}\varepsilon_{UM}Z_i$ We thus used only two sources of errors that were simulated using two multiplicative parameters, one with uncertainty shared by all cohort members, and the other varied at random for individuals in the cohort. Under this simplification SUMA generates uncertainties corresponding to Berkson type errors, i.e. the true values vary about the mean at random [1]. We let $\sigma^2_{SM} = Var(\varepsilon_{SM})$ and $\sigma^2_{UM} = Var(\varepsilon_{UM})$ and examined the impact of different magnitudes of exposure uncertainty on interval estimation using four scenarios with different values for $\sigma^2_{SM}$ and $\sigma^2_{UM}$, summarized in Table 3. The first scenario uses the same values as used in [4], and the other scenarios had twice and three times the magnitude of the shared multiplicative error.

For each scenario we first generated 5,000 realizations of a possibly true cohort dose vector with sample size of N = 1,000 individuals in the cohort. Fig 3 illustrates the degree of uncertainty for each scenario by plotting the cumulative distribution function (CDF) of the study doses. The thick black line represents the single vector of individual mean doses, i.e., the vector of individual doses averaged per each person across all dose realizations. Each thin gray line represents one of the 5,000 realizations of the cohort dose vector per test scenario. The width of the gray area in the plots represents the degree of shared uncertainty in the dose distributions, where a wider gray area corresponds to a larger shared uncertainty component. As can be seen, each scenario represents different degrees of shared uncertainty.

In this exercise, one of the generated dose vectors was assumed to be the "true dose" to generate the disease outcome vectors $\underline{Y}$ from the Poisson regression model described in Table 4 with true values of β = 0 and of β = 3 for the dose-response regression coefficient. Age was generated using a uniform distribution with support 20 to 50, and gender was generated using a Bernoulli distribution with p = 0.5 to obtain equal proportions of men and women. The person-years for a stratum were generated from an exponential distribution with rate = 0.5. In the model fitting step, we excluded the specific cohort dose vector used to generate the disease out-

**Table 3. Variances used for data generation in the simulation study based on the SUMA model.** $\sigma^2_{SM}$ denotes the variance of the shared multiplicative error ($\varepsilon_{SM}$), and $\sigma^2_M$ the variance of the unshared multiplicative error ($\varepsilon_{UM,i}$). See Fig 3 for a graphical depictions of scenario realizations.

| Scenario | $\sigma^2_{SM}$ | $\sigma^2_{UM}$ | $\sigma^2_{SM}/\sigma^2_{UM}$ | GSD (SM) | GSD(UM) | GSD(SM)/GSD(UM) |
|----------|------------------|------------------|-------------------------------|----------|---------|------------------|
| 1 | 0.318 | 0.223 | 1.43 | 1.8 | 1.6 | 1.1 |
| 2 | 0.632 | 0.223 | 2.83 | 2.2 | 1.6 | 1.4 |
| 3 | 0.632 | 0.446 | 1.42 | 2.2 | 2.0 | 1.1 |
| 4 | 0.954 | 0.223 | 4.28 | 2.7 | 1.6 | 1.7 |

Abbreviations: SM: shared multiplicative error; UM: unshared multiplicative error; GSD: geometric standard deviation

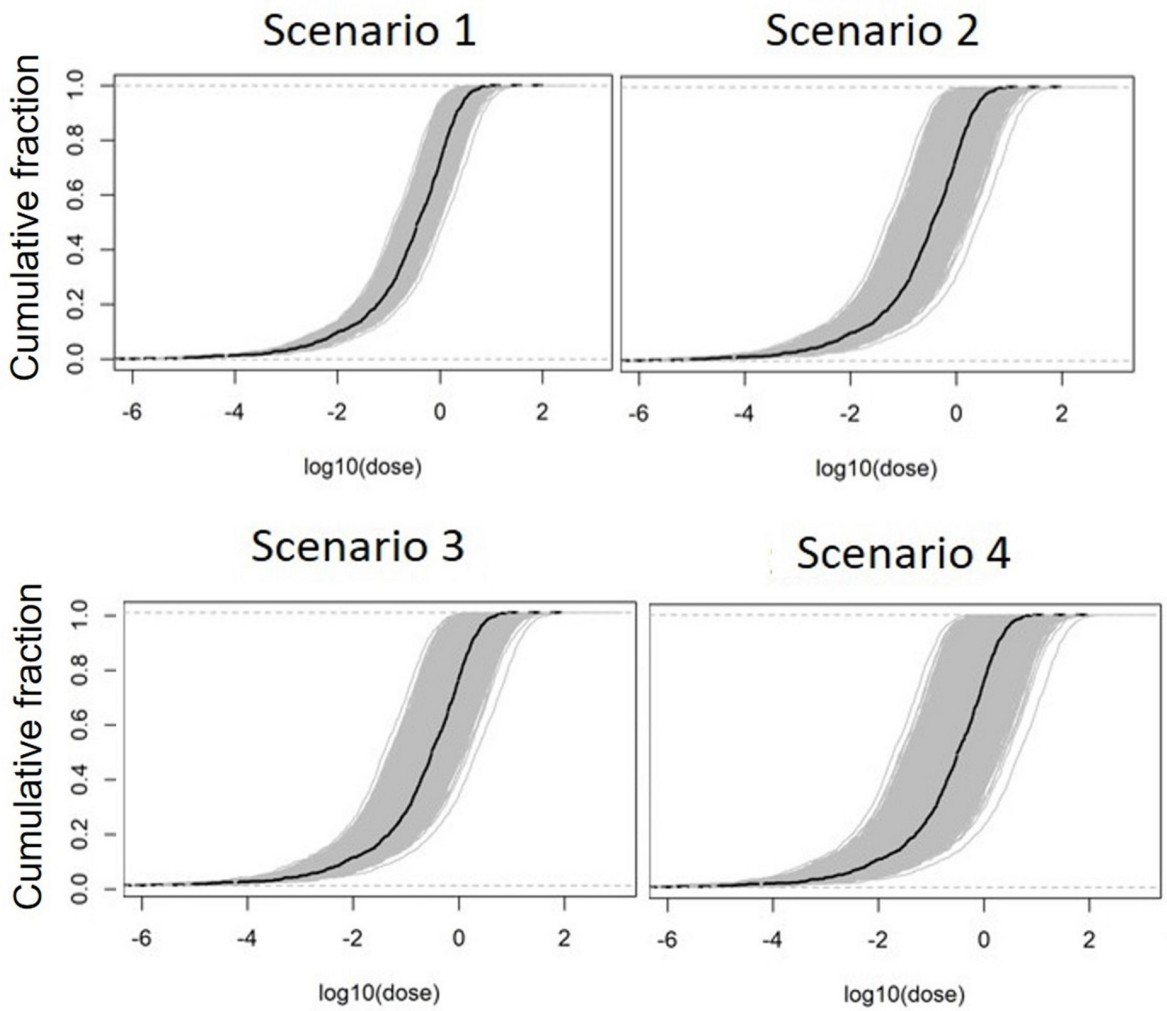

**Fig 3. Cumulative distribution function plots of doses for the four scenarios with increasing uncertainty corresponding to the settings in Tables 1 and 2.**

comes to reflect a realistic setting for analysis. Thus, for each test 4,999 realizations were evaluated for their goodness of fit to the dose response for both CIM and FMA.

We summarize the simulation results using the same operating characteristics we used in Section 3. In addition to β = 3, we also evaluated the operating characteristics for the possibility of no association of dose with outcome (β = 0).

## Simulation results

We first generated 5,000 different count outcome vectors for individual disease status in the cohort under the null model (β = 0) using doses generated under Scenario 1 in Table 3, and examined LER, CR, RER, absolute bias (AB), LCI, and the upper limit of 95% confidence interval (UL). As shown in Table 5, both CIM and FMA had similar operating characteristics and the coverage rates of the corresponding 95% CIs were close to the nominal rate of 95%.

Table 6 shows CR, LER, RER, LCI, ARB, and RUL when disease outcome data were generated from a Poisson model with β = 3 for doses generated under all settings in Table 4. Here

**Table 4. Model for simulating cohort disease vectors.**

| Outcome; Statistical model | Model |
|---|---|
| Count outcome Y; Poisson regression | $y_i \sim Poisson(\lambda_i \times PY_i)$ where $\lambda_i = exp\left(1 + 1.5 * Gender_i + 2 * log\left(\frac{Age_i}{40}\right)^2\right)\left(1 + \beta * dose_i\right)$ and $PY_i$ is person-years for i$^{th}$ stratum |

both CIM and FMA also had empirical CRs close to the nominal 95% level for all scenarios. The averages of the coverage rates of FMA and CIM based CIs were about 95.7% and 96.2%, respectively. The two tail error rates of CIM were similar, but FMA had a higher LER than RER. That suggests that FMA is more likely to underestimate β, but less likely to overestimate β. CIM had slightly smaller absolute relative bias (ARB,) than FMA.

As shown in Table 6, when doses were generated under Scenario 3, CIM had an almost 31% larger average LCI compared to FMA (8.18 vs 6.22) and 60% larger RULs (3.12 vs 1.95). While CIM and FMA based CIs had similar coverage rates, around 95%, those obtained from FMA were much shorter in length with a lower upper limit. Thus the 95% CIs from FMA were more informative than those obtained from CIM when the magnitude of systematic and/or shared uncertainty was large. For large systematic and/or shared uncertainty in dose estimation, the length of the CIs for CIM increased substantially. For easy visualization in comparing CIM and FMA for each scenario, 100 tests were randomly selected from 5,000 tests and were rearranged by ERR(D) estimate of CIM (Fig 4, with CIM in left column and FMA results in the right column). Most FMA upper limits of the 95% CIs were smaller than those of CIM. In Scenario 1, 87% of the upper limits of FMA based CIs were smaller than those of the CIM based CIs. For other scenarios, around 95% of the upper limits of FMA based CI were smaller than those of the CIM based Cis.

## Discussion

In this paper, we propose a computationally efficient and relatively simple novel frequentist model averaging approach (FMA) to accommodate multiple exposure realizations for the analysis of exposure-response relationships. For the linear excess odds ratio (EOR) model with a binary outcome we used in our real data analysis and the simulation study, the prior for the BMA approach for the EOR/Gy, β, was a truncated normal distribution, truncated at zero (i.e., non-negative constraint). This ensures that $1 + \beta D_i > 0$ in the function $log[1 + \beta D_i]$, that is part of the EOR model. Although this non-negativity constraint for the EOR estimation in the BMA approach introduced differences in the lower limit of 95% CIs, the discrepancy of the lower bound was negligible for small or moderate estimates of EOR/Gy (see Table 2). We show that FMA provides association and uncertainty estimates comparable to those of the

**Table 5. Performance of CIM and FMA for outcomes generated from a Poisson regression with β = 0 (CR: 95% CI coverage rate, LER: left error rate, RER: right error rate, LCI: length of 95% CIs, AB: absolute bias, UL: upper limit) for the null model (β = 0) using doses generated under Scenario 1.** Results are summarized from 5,000 tests with sample size of 1,000 using different pre-selected 2DMC realizations of a cohort dose vector for each of the 5,000 tests. CR, RER, and LER were defined as the percentage of times the true value of the slope of the null dose response (β = 0) was inside the 95% CI, and below the lower limit, or above the upper limit, respectively.

| Performance measure | CIM | FMA |
|---|---|---|
| Coverage Rate (%) (LER, RER) | 93.4 (5.90, 0.73) | 95.9 (3.83, 0.23) |
| AB Mean (95%CI) | 0.04 (0.001, 0.104) | 0.04 (0.002, 0.111) |
| LCI Mean (95%CI) | 0.20 (0.16, 0.29) | 0.21 (0.16, 0.30) |
| UL Mean (95%CI) | 0.10 (-0.018, 0.275) | 0.11 (-0.007, 0.268) |

**Table 6. A comparison of CIM and FMA estimates for the slope of a linear dose response (ERR/Gy) for outcomes generated from a Poisson model (CR: 95% CI coverage rate, LER: left error rate, RER: right error rate, LCI: length of 95% CIs, ARB: absolute relative bias, RUL: relative upper limit).** CR, RER, and LER were defined as the percentage of times the true value of the slope of the dose response was inside the CI, below the lower limit, and above the upper limit, respectively. ARB was calculated as the absolute difference between the estimated value of the slope of a linear dose response and the pre-determined true value ($\beta = 3$) divided by the true value of the slope.

| | Simulation Scenario | | | | | | | |
|---|---|---|---|---|---|---|---|---|
| | 1 | | 2 | | 3 | | 4 | |
| | CIM | FMA | CIM | FMA | CIM | FMA | CIM | FMA |
| CR (%) (LER, RER) | 96.7 (1.72, 1.54) | 96.5 (2.92, 0.6) | 96.1 (2.0, 1.92) | 96 (3.3, 0.66) | 95.8 (2.18, 1.98) | 94.4 (5.38, 0.2) | 96.1 (2.06, 1.88) | 96 (3.72, 0.28) |
| ARB Mean (95% CI) | 0.23 (0.008, 0.675) | 0.25 (0.008, 0.85) | 0.39 (0.014, 1.325) | 0.55 (0.017, 2.251) | 0.40 (0.017, 1.358) | 0.47 (0.018, 1.90) | 0.53 (0.026, 1.94) | 0.93 (0.025, 4.15) |
| LCI Mean (95% CI) | 3.96 (2.20, 6.65) | 3.71 (2.10, 6.46) | 7.98 (2.71, 18.5) | 7.04 (2.49, 16.3) | 8.18 (2.73, 19.4) | 6.22 (2.24, 14.4) | 13.77 (2.99, 40.6) | 11.17 (2.55, 31.9) |
| RUL Mean (95% CI) | 1.88 (1.88, 1.89) | 1.61 (1.46, 1.82) | 3.07 (3.06, 3.10) | 1.97 (1.66, 2.35) | 3.12 (3.11, 3.16) | 1.95 (1.61, 2.42) | 4.92 (4.90, 4.98) | 2.29 (1.86, 2.81) |

computationally much more burdensome BMA. In a simulation study, we compared the operating characteristics of the FMA to CIM, which also accommodates multiple dose realizations in the computation of the confidence intervals of the association estimates. We used a simple two parameter SUMA algorithm for generating exposure realizations assuming all shared and unshared uncertainties were random and multiplicative, and centered on the single cohort vector of individual mean doses, essentially implying a Berkson error model, where the cohort vector of individual mean dose is an unbiased surrogate for the true dose vector. While SUMA is easy to use for simulations, the 2DMC method makes no assumptions that the vector of mean doses per person is unbiased. Thus, these simulation results have to be interpreted with caution due to the simplified structure inherent in the two-parameter SUMA method.

As with regression calibration and conventional regression, CIM uses the single cohort vector of individual mean doses (averaged over all multiple realizations for each cohort member) for estimating the dose-response association parameter. This results in CIM estimates with a relatively small absolute bias when averaged across all tests. Different properties of the estimated confidence intervals between CIM and FMA come from their differences in aggregating the *K* different models. CIM does not distinguish dose vectors that yield models with a good fit to the cohort outcomes and models with poor fit in the estimation of the variance of $\beta$. In addition, CIM assumes that the single vector of individual mean doses is a good approximation to the unknown true dose. This assumption is likely unrealistic. In most reconstructions of past radiation exposure, there is a substantial lack-of-knowledge about the exposure conditions and the parameters of the dose estimation model are associated with large and complex amounts of systematic and/or shared uncertainty.

FMA weighs the estimates of $\beta$ obtained for different dose vectors by their goodness of fit to the disease outcome data. If a particular model weight is very small (e.g., less than 0.001), then the corresponding model contributes minimally to the estimate of $\beta$ and its confidence interval. This is reflected in FMA yielding shorter confidence intervals for the estimates of $\beta$ than CIM. In Fig 5, FMA weights were displayed for selected three tests for each scenario. Among the 4,999 realizations of possibly true cohort dose vectors, only a small proportion had goodness-of-fit weights of impactful magnitude.

Previously, Kwon et al. [3] proposed a Bayesian model averaging (BMA) approach to estimate dose-response parameters while accommodating shared uncertainties in exposures. We proposed FMA as a computationally more efficient alternative to BMA that is also much easier to implement using available statistical software such as SAS, R, or EPICURE [31–33]. For

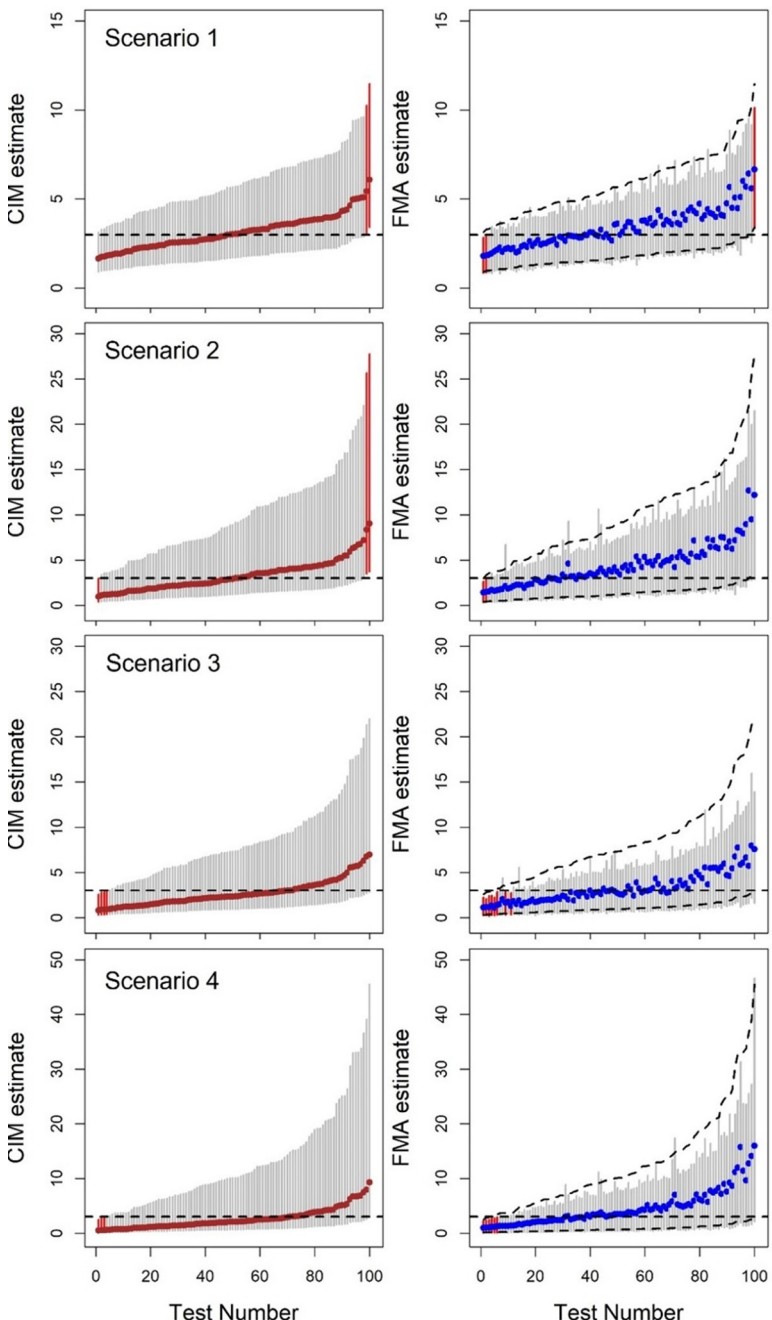

**Fig 4.** Comparison between CIM (left column) and FMA (right column) using 100 selected tests for the four scenarios with increasing uncertainty. Red vertical line denotes 95%CI does not include the true value. Upward dashed line on the right column indicates for lower and upper limits of CIM for easy comparison. Dashed horizontal line denotes the value of true ERR/Gy (= 3).

FMA, given an exposure realization, any conventional dose-response analysis can be performed that yields estimates of the regression coefficients for the response function, corresponding standard errors, and goodness-of-fit measures (e.g., AIC, BIC, DIC) [28] which can then be aggregated.

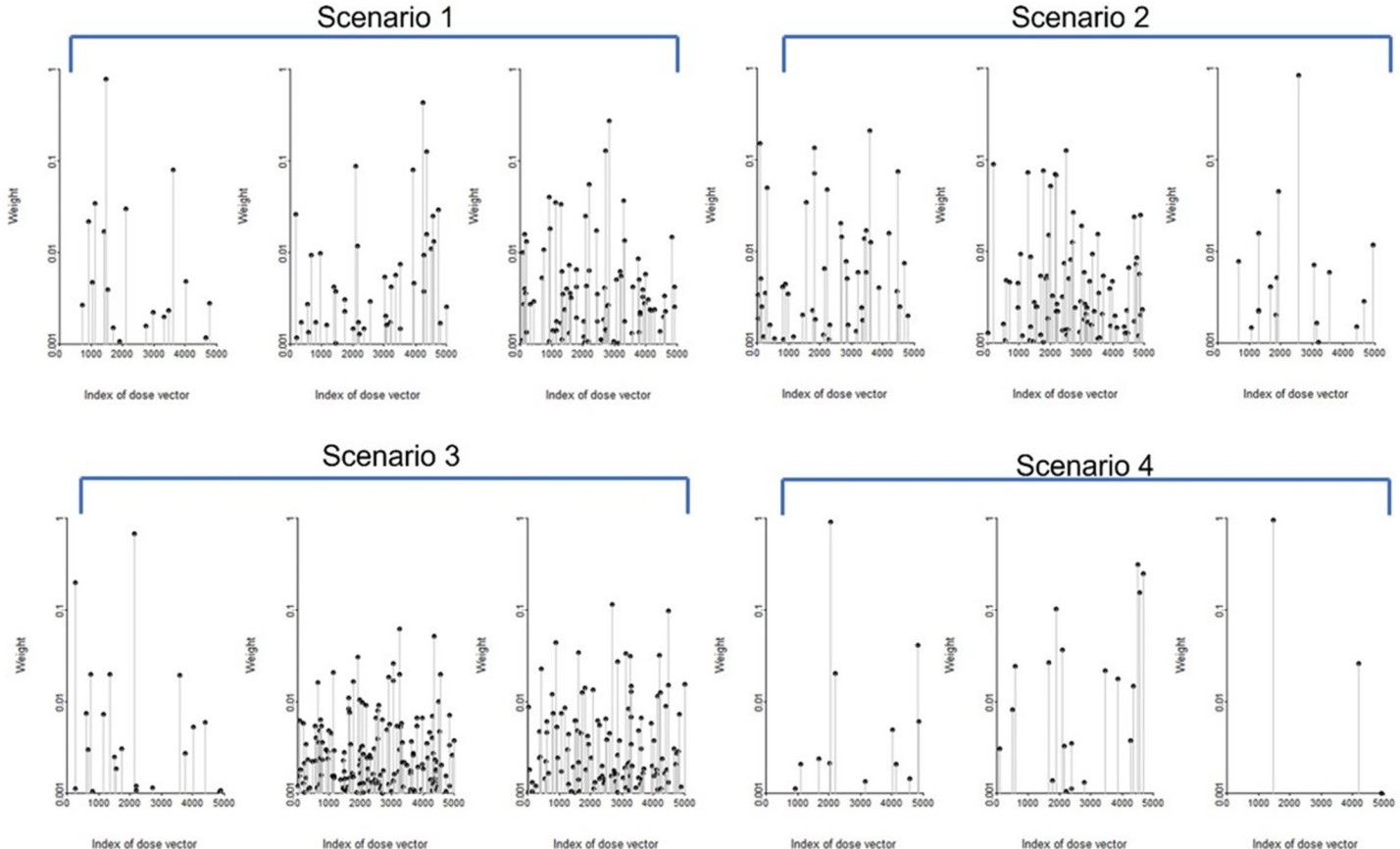

**Fig 5. FMA weight plots for selected tests for Scenario 1, Scenario 2, Scenario 3, and Scenario 4.**

In our simulations based on real data we found that FMA and BMA had very similar performance characteristics.

Another limitation of BMA is that it is computationally challenging for very large datasets. For example, for analyzing risk in the EPI-CT study, in which radiation-induced cancer risk from pediatric computed tomography (CT) scans was assessed for over 900,000 individuals [34], the use of BMA would be a daunting task and require unrealistically large computational resources, while the use of FMA appears eminently feasible. Another computational advantage of FMA is that the implementation can easily be parallelized. This is impossible for both BMA and CIM since all realizations are considered simultaneously in these two approaches. Thus, FMA provides a viable computational approach with considerable advantages for the analysis of very large datasets with multiple exposure realization.

### Appendix: Corrected information matrix (CIM) method

The CIM method [4] estimates the regression coefficients for the dose-response function by using the mean dose, $Z = E(X|W)$, in model (1) instead of $X$, where W denotes all available information about dose, e.g., years of residency, age at exposure and sex. For individual $i$ in the cohort, $E(X|W)$ is estimated by the empirical mean of $X_{i1}, \ldots, X_{ik}$, the corresponding dose realizations. The key (untestable and very strong) assumption for the CIM approach is that the conditional mean $Z = E(X|W)$ is an unbiased estimate of the individuals' true dose X.

To accommodate exposure uncertainty in the confidence intervals for the regression coefficients when using Z instead of individual doses X, a correction term for the variance that is a function of $Cov(X|Z)$ is used, as described here in brief. Based on a Taylor expansion of the score function $S_Z$ corresponding to the log-likelihood in (6) around the true vector of parameter values $\Theta$, we obtain $\hat{\Theta} - \Theta \approx I_Z^{-1} S_Z$, where $I_Z$ is the Fisher information. Both, $I_Z^{-1}$ and $S_Z$, are estimated using the mean dose, Z. From the Taylor expansion we also have that $Var(\hat{\Theta}) \approx I_Z^{-1} Var(S_Z) I_Z^{-1}$, where $Var(S_Z) = I_Z + \beta^2 QG Cov(X \mid Z) GQ'$, and $Q = [Q_1, \ldots, Q_N]$ with

$$Q_i = \begin{bmatrix} 1 \\ C_i \\ \dfrac{Z_i}{1 + \beta Z_i} \end{bmatrix}, i = 1, \ldots, N,$$

and $G = diag[exp(\alpha_0 + \alpha_1 C), exp(\alpha_0 + \alpha_1 C), exp(\alpha_0 + \alpha_1 C)]$. Letting $M = QG$, the covariance matrix of the vector of parameter estimates is

$$Var(\hat{\Theta}) = I_Z^{-1} + \beta^2 I_Z^{-1} M Cov(X \mid Z) M' I_Z^{-1}. \tag{A1}$$

A Wald-type corrected $\alpha$ level confidence interval for the ERR parameter $\beta$ in (5) is then given by

$$\hat{\beta} \pm z_{(1-\alpha/2)} \sqrt{\left( I_Z^{-1} + \hat{\beta}^2 I_Z^{-1} M Cov(X \mid Z) M' I_Z^{-1} \right)_{\beta,\beta}}, \tag{A2}$$

where $z_{(1-\alpha/2)}$ denotes the $1 - \alpha/2$ quantile of the standard normal distribution. A limitation of the CIM approach is that it can be applied only to linear and Poisson regression models, but not logistic or Cox proportional hazards models. To fit Poisson models to large datasets, CIM requires aggregating the data into bins of individuals of the same confounder, effect modifier and dose combinations to make computations feasible.

## Acknowledgments

We thank the two referees and the Academic Editor for their valuable comments that substantially improved this paper.

## Author Contributions

**Conceptualization:** Deukwoo Kwon, Steven L. Simon, F. Owen Hoffman, Ruth M. Pfeiffer.

**Data curation:** Deukwoo Kwon.

**Formal analysis:** Deukwoo Kwon.

**Investigation:** Deukwoo Kwon.

**Methodology:** Deukwoo Kwon, Ruth M. Pfeiffer.

**Software:** Deukwoo Kwon.

**Visualization:** Deukwoo Kwon.

**Writing – original draft:** Deukwoo Kwon.

**Writing – review & editing:** Deukwoo Kwon, Steven L. Simon, F. Owen Hoffman, Ruth M. Pfeiffer.

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
