## [Decision Letter · Decision Letter 0]

22 Jun 2023

PONE-D-23-15137Frequentist model averaging for analysis of dose–response in epidemiologic studies with complex exposure uncertaintyPLOS ONE

Dear Dr. Kwon,

Thank you for submitting your manuscript to PLOS ONE. After careful consideration, we feel that it has merit but does not fully meet PLOS ONE’s publication criteria as it currently stands. Therefore, we invite you to submit a revised version of the manuscript that addresses the points raised during the review process.

We look forward to receiving your revised manuscript.

Kind regards,

Viacheslav Kovtun, Dr.Sc., Ph.D.

Academic Editor

PLOS ONE

Journal Requirements:

2. Please note that PLOS ONE has specific guidelines on code sharing for submissions in which author-generated code underpins the findings in the manuscript. In these cases, all author-generated code must be made available without restrictions upon publication of the work. 

Please review our guidelines at https://journals.plos.org/plosone/s/materials-and-software-sharing#loc-sharing-code and ensure that your code is shared in a way that follows best practice and facilitates reproducibility and reuse.

"DK was partially supported by the support provided by the Biostatistics/ Epidemiology/ Research Design (BERD) component of the Center for Clinical and Translational Sciences (CCTS) for this project that is currently funded through a grant (UL1TR003167), funded by the National Center for Advancing Translational Sciences (NCATS), awarded to the University of Texas Health Science Center at Houston."

7. Please include a separate caption for each figure in your manuscript.

Reviewers' comments:

Reviewer's Responses to Questions

**Comments to the Author**

1. Is the manuscript technically sound, and do the data support the conclusions?

Reviewer #1: Yes

Reviewer #2: Yes

2. Has the statistical analysis been performed appropriately and rigorously? 

Reviewer #1: Yes

Reviewer #2: Yes

3. Have the authors made all data underlying the findings in their manuscript fully available?

Reviewer #1: No

Reviewer #2: Yes

4. Is the manuscript presented in an intelligible fashion and written in standard English?

Reviewer #1: Yes

Reviewer #2: Yes

5. Review Comments to the Author

Reviewer #1: Manuscript concisely and cogently describes the application of model averaging, specifically frequentist model averaging (FMA), to exposure epidemiology studies and compares against existing Bayesian model averaging. The use of FMA in exposure epidemiology studies appears to be novel and this detailed description along with comparisons using simulations and actual data from an existing study are important contributions to the literature. Although the paper is well written, there are several areas where additional clarification or explanation would strengthen the presentation.

1. Page 2 - "...all the analytical advantages of the BMA..." Prior to this sentence, the analytical advantages of BMA should be explicitly described for readers.

2. Page 2 - "...for association parameters that have proper coverage..." the use of 'proper' is problematic here as it is a relative term, better to describe the statistical implications of the coverage specifically. (How does one know which is more proper, wider coverage?, narrower coverage?...).

3. Page 2 - "In simulations, the FMA CIs are also shorter than those of ...". 'shorter' seems out of place here, does this mean faster computationally, or lower? Clarify.

4. Page 2 - The use of the term 'expensive' here up front should be more explicitly stated so that when it is utilized later in the manuscript, it will enable a clearer understanding of the implication.

5. Page 2 - "Traditional statistical methods for measurement-error correction..." Rather than 'correction' perhaps 'uncertainty incorporation' or similar.

6. Page 4 and throughout - rather than use 'systematic/shared' perhaps 'systematic and/or shared'.

7. Page 4, 2nd para, when FMA is introduced in more detail - there are many references in the literature on FMA approaches, several important general descriptions should likely be cited. It is noted that none appear in the current reference list. While application of FMA to epidemiology study may be novel, the methodology has had wide application in the more general literature.

8. Page 8 - "...for simulating multiple realizations of dose vectors is the SUMA..." , 'is' should be 'known as'

9. Page 12 - much more should be said about the 'weakly informative multinomial distribution' that employs priors and hyperpriors. Why weakly informative? And if so, why incorporate at all (making system more efficient). What is implication if not applied? How are decisions about the priors and hyperpriors made?

10. Page 17 - the discrepancy of the lower limits of the 95% CIs (stated as due to the non-negativity constraint for the EOR in the BMA approach) should be addressed in the discussion - implications?

11. Page 21 - Discussion. 2nd sentence. perhaps a place to add "...although the non-negative constraint on EOR inherently employed by BMA introduced differences in lower limit of CIs." or similar acknowledgement.

12. Figure 5 - remove the commentary notes from the figure caption.

13. Table 1A - as noted in the spurious commentary note that still exists in the table...Rather than use these relative terms 'lower' vs 'higher', use category descriptors that explicitly state the parameter characteristic.

Reviewer #2: The authors, in this paper, proposed a novel frequentist model averaging (FMA) approach which has all the analytical advantages of the Bayesian model averaging method. The paper involves an interesting idea but it requires some clarifications:

1. What are prime parameters of the problem?

2. Is there any base to consider the comparison between FMA and BMA.

3. You have correlated your work with the experimental one using “true dose”. What is the fact behind this scenario.

4. what are the average coverage rates of all the supposed methods e.g. FMA, BMA CIM etc.

5. Mention main findings of the paper in a separate section (e.g. Conclusions) at the end of the paper.

6. PLOS authors have the option to publish the peer review history of their article (what does this mean?). If published, this will include your full peer review and any attached files.

Reviewer #1: No

Reviewer #2: **Yes: **Sohail Ahmad

---

## [Author Response · Author response to Decision Letter 0]

21 Jul 2023

Responses to Reviewers’ comments

Reviewer #1: 

Manuscript concisely and cogently describes the application of model averaging, specifically frequentist model averaging (FMA), to exposure epidemiology studies and compares against existing Bayesian model averaging. The use of FMA in exposure epidemiology studies appears to be novel and this detailed description along with comparisons using simulations and actual data from an existing study are important contributions to the literature. Although the paper is well written, there are several areas where additional clarification or explanation would strengthen the presentation.

1. Page 2 - "...all the analytical advantages of the BMA..." Prior to this sentence, the analytical advantages of BMA should be explicitly described for readers.

Thank you for the comment: We added the following sentences on page 2.

“The conventional approach uses mean doses in the dose-exposure analysis and this approach does not fully account for exposure uncertainty. The traditional measurement-error model approach considers limited exposure uncertainty in the dose-exposure analysis. … The analytic advantage of the BMA is to incorporate exposure uncertainty more completely into risk estimation.” 

2. Page 2 - "...for association parameters that have proper coverage..." the use of 'proper' is problematic here as it is a relative term, better to describe the statistical implications of the coverage specifically. (How does one know which is more proper, wider coverage?, narrower coverage?...).

We replaced this sentence “We show in simulations that, like BMA, FMA yields confidence/credibility intervals (CIs) for association parameters that have proper coverage and thus allow for valid inference.” with “We show in simulations that, like BMA, FMA yields 95% confidence intervals for association parameters that have close to 95% coverage rate.”

3. Page 2 - "In simulations, the FMA CIs are also shorter than those of ...". 'shorter' seems out of place here, does this mean faster computationally, or lower? Clarify.

We replaced this sentence “In simulations, the FMA CIs are also shorter than those of another frequentist statistical procedure to incorporate exposure uncertainty in the CI calculation, the corrected information matrix (CIM) method, which is computationally similarly as expensive as BMA for large datasets.” with “In simulations, the FMA produces CIs that are shorter than those of another frequentist approach, the corrected information matrix (CIM) method.”

4. Page 2 - The use of the term 'expensive' here up front should be more explicitly stated so that when it is utilized later in the manuscript, it will enable a clearer understanding of the implication.

Thank you for your comment. Due to the word limit in Abstract, we removed this and added some sentences in the section for FMA (see pages 13 and 15). 

“A drawback of the BMA approach is that it is extremely expensive computationally with large dataset. In implementing the BMA, the whole dataset has to be read into memory and a large model space needs to be evaluated to obtain posterior model probabilities. For example, it takes several days on a personal computer with a multicore processor and 24GB memory to get the BMA result for the example data in this paper using traditional Bayesian computational softwares such as WinBUGS [22] and JAGS [23].”

“As FMA allows for parallel implementations there are huge savings in computational time when the sample size, N, and the number of exposure realizations, K, are large. To run BMA, the whole dataset has to be read into computer memory space, which typically prevents BMA analyses on standard single computers with large datasets.”

5. Page 2 - "Traditional statistical methods for measurement-error correction..." Rather than 'correction' perhaps 'uncertainty incorporation' or similar.

We changed the sentence as suggested.

6. Page 4 and throughout - rather than use 'systematic/shared' perhaps 'systematic and/or shared'.

We made the suggested change.

7. Page 4, 2nd para, when FMA is introduced in more detail - there are many references in the literature on FMA approaches, several important general descriptions should likely be cited. It is noted that none appear in the current reference list. While application of FMA to epidemiology study may be novel, the methodology has had wide application in the more general literature.

Thank you for your comments. We have now added a more general overview and description of the FMA and added several references on FMA to Section 2.4.2.

8. Page 8 - "...for simulating multiple realizations of dose vectors is the SUMA..." , 'is' should be 'known as'

Thank you for your careful reading. We have corrected this error. 

9. Page 12 - much more should be said about the 'weakly informative multinomial distribution' that employs priors and hyperpriors. Why weakly informative? And if so, why incorporate at all (making system more efficient). What is implication if not applied? How are decisions about the priors and hyperpriors made?

Thank you for your comment. We added the following paragraph to clarify the rationale for using weakly informative priors: 

“Using weakly informative or non-informative priors in the Bayesian analysis reduces the impact of the prior on the inference and thus maximizes the influence of data. The weakly informative prior for the dose index parameter, �, reflects our lack of knowledge about which dose realization is closest to the true unknown dose, and sets all dose realization to be equally likely to be closest to the true dose. As � is a categorical variable, its prior is usually a Dirichlet prior with parameters all equal to one.” 

The weakly informative priors for the regression coefficients are normal distributions with large variances (i.e., 100 or 1,000), which makes the shape of the corresponding densities almost flat and thus uninformative. 

10. Page 17 - the discrepancy of the lower limits of the 95% CIs (stated as due to the non-negativity constraint for the EOR in the BMA approach) should be addressed in the discussion - implications?

We added some further discussion to the paper. Please also see our response to comment 11 below.

11. Page 21 - Discussion. 2nd sentence. perhaps a place to add "...although the non-negative constraint on EOR inherently employed by BMA introduced differences in lower limit of CIs." or similar acknowledgement.

Thank you for the suggestion. We added the following sentences:

“For the linear excess odds ratio (EOR) model with a binary outcome we used in our real data analysis and the simulation study, the prior for the BMA approach for the EOR/Gy, β, was a truncated normal distribution, truncated at zero (i.e., non-negative constraint). This ensures that 1+βD_i>0 in the function log[1+βD_i ], that is part of the EOR model. Although this non-negativity constraint for the EOR estimation in the BMA approach introduced differences in the lower limit of 95% CIs, the discrepancy of the lower bound was negligible for small or moderate estimates of EOR/Gy (see Table 1B).”

12. Figure 5 - remove the commentary notes from the figure caption.

Thank you for pointing out this mistake. We removed it.

13. Table 1A - as noted in the spurious commentary note that still exists in the table...Rather than use these relative terms 'lower' vs 'higher', use category descriptors that explicitly state the parameter characteristic.

Thank you for pointing out our mistake. We removed it.

 

Reviewer #2: 

The authors, in this paper, proposed a novel frequentist model averaging (FMA) approach which has all the analytical advantages of the Bayesian model averaging method. The paper involves an interesting idea but it requires some clarifications:

1. What are prime parameters of the problem?

Our main parameter of interest is EOR/Gy, β. We added the following sentence to page 11. “The primary parameter of interest is β, which captures the dose-response relationship”

2. Is there any base to consider the comparison between FMA and BMA.

Both these methods use the idea of model averaging to account for exposure uncertainty. Thus we see no compelling reason not to compare their ability to estimate the parameter of interest and quantify its uncertainty. However, FMA takes a frequentist perspective while BMA tackles the problem from a Bayesian viewpoint. We proposed FMA as a computationally more efficient alternative to BMA and thus is it important to show that it does not produce less efficient inference than BMA. 

3. You have correlated your work with the experimental one using “true dose”. What is the fact behind this scenario.

In some occupational and environmental epidemiologic study, we need to deal with large uncertainty in exposure measurement and knowledge about the true exposure value is unavailable. In the simulation studies, we mimicked this realistic situation. However, to generate disease outcomes we did assume a particular value for each person to be the “true exposure value” and then removed this true exposure values in the analysis stage. 

4. what are the average coverage rates of all the supposed methods e.g. FMA, BMA CIM etc.

We apologize for the lack of clarity and added the following sentences.

“To compare the efficiency of BMA and FMA, we present the averages of their respective coverage rates, which were very similar, about 97.1% (BMA) and 98.4% (FMA) for the setting of small uncertainty; and 95.8% (BMA) and 96.7% (FMA) for the setting of large uncertainty.” (page 18)

“The averages of the coverage rates of FMA and CIM based CIs were about 95.7% (FMA) and 96.2% (CIM).” (page 22)

5. Mention main findings of the paper in a separate section (e.g. Conclusions) at the end of the paper.

We revised the last paragraph of the paper on pages 25 and 26. 

“Previously, Kwon et al. [3] proposed a Bayesian model averaging (BMA) approach to estimate dose-response parameters while accommodating shared uncertainties in exposures. We proposed FMA as a computationally more efficient alternative to BMA that is also much easier to implement using available statistical software such as SAS, R, or EPICURE [25-27]. For FMA, given an exposure realization, any conventional dose-response analysis can be performed that yields estimates of the regression coefficients for the response function, corresponding standard errors, and goodness-of-fit measures (e.g., AIC, BIC, DIC) [28] which can then be aggregated. 

In our simulations based on real data we found that FMA and BMA had very similar performance characteristics. 

Another limitation of BMA is that it is computationally challenging for very large datasets. For example, for analyzing risk in the EPI-CT study, in which radiation-induced cancer risk from pediatric computed tomography (CT) scans was assessed for over 900,000 individuals [24], the use of BMA would be a daunting task and require unrealistically large computational resources, while the use of FMA appears eminently feasible. Another computational advantage of FMA is that the implementation can easily be parallelized. This is impossible for both BMA and CIM since all realizations are considered simultaneously in these two approaches. Thus, FMA provides a viable computational approach with considerable advantages for the analysis of very large datasets.”

---

## [Decision Letter · Decision Letter 1]

10 Aug 2023

Frequentist model averaging for analysis of dose–response in epidemiologic studies with complex exposure uncertainty

PONE-D-23-15137R1

Dear Dr. Kwon,

We’re pleased to inform you that your manuscript has been judged scientifically suitable for publication and will be formally accepted for publication once it meets all outstanding technical requirements.

Kind regards,

Viacheslav Kovtun, Dr.Sc., Ph.D.

Academic Editor

PLOS ONE

Additional Editor Comments (optional):

Reviewers' comments:

Reviewer's Responses to Questions

**Comments to the Author**

1. If the authors have adequately addressed your comments raised in a previous round of review and you feel that this manuscript is now acceptable for publication, you may indicate that here to bypass the “Comments to the Author” section, enter your conflict of interest statement in the “Confidential to Editor” section, and submit your "Accept" recommendation.

Reviewer #1: All comments have been addressed

Reviewer #2: All comments have been addressed

2. Is the manuscript technically sound, and do the data support the conclusions?

Reviewer #1: Yes

Reviewer #2: Yes

3. Has the statistical analysis been performed appropriately and rigorously? 

Reviewer #1: Yes

Reviewer #2: Yes

4. Have the authors made all data underlying the findings in their manuscript fully available?

Reviewer #1: Yes

Reviewer #2: (No Response)

5. Is the manuscript presented in an intelligible fashion and written in standard English?

Reviewer #1: Yes

Reviewer #2: Yes

6. Review Comments to the Author

Reviewer #1: All comments have been adequately addressed. This improved manuscript is acceptable for publication.

Reviewer #2: The manuscript has been revised significantly. So I accept the revised version of the paper for publication.

7. PLOS authors have the option to publish the peer review history of their article (what does this mean?). If published, this will include your full peer review and any attached files.

Reviewer #1: No

Reviewer #2: **Yes: **Sohail Ahmad

---

## [Editor Report · Acceptance letter]

24 Aug 2023

PONE-D-23-15137R1 

Frequentist model averaging for analysis of dose–response in epidemiologic studies with complex exposure uncertainty 

Dear Dr. Kwon:

I'm pleased to inform you that your manuscript has been deemed suitable for publication in PLOS ONE. Congratulations! Your manuscript is now with our production department. 

Kind regards, 

on behalf of

Professor Viacheslav Kovtun 

Academic Editor

PLOS ONE